# Nurses’ Adaptations in Caring for COVID-19 Patients: A Grounded Theory Study

**DOI:** 10.3390/ijerph181910141

**Published:** 2021-09-27

**Authors:** Jinhee Kim, Suhyun Kim

**Affiliations:** 1Department of Nursing, College of Medicine, Chosun University, 309 Pilmun-daero, Dong-Gu, Gwangju 61452, Korea; jinheeara@chosun.ac.kr; 2Department of Nursing, Nambu University, 23 Cheomdanjungang-ro, Gwangsan-Gu, Gwangju 62271, Korea

**Keywords:** nurse, COVID-19, disease outbreak, qualitative research

## Abstract

As the novel coronavirus (COVID-19) pandemic continues, frontline nurses caring for COVID-19 patients are experiencing severe fatigue and mental stress. This study explored nurses’ adaptation process in caring for COVID-19 patients and examined how nurses interact with the phenomenon using a grounded theory approach. The study aimed to develop a substantive theory and provide basic data with which to develop intervention programs that can support nurses caring for COVID-19 patients. Data were collected between 7 August and 31 October 2020, via face-to-face in-depth interviews with 23 nurses who had been caring for COVID-19 patients for six months or more at a nationally designated COVID-19 hospital. Sampling was started purposively and continued theoretically. Data analysis, performed using the method proposed by Strauss and Corbin, resulted in 13 main categories, the core one being “growing as a proficient nurse alongside comrades on the COVID-19 frontline”. The study’s results identify the nurses’ adaptation process in caring for COVID-19 patients and their reactions to the circumstances around it. Ensuring that nurses can systematically cope with emerging infectious diseases requires regularly providing them with basic education on caring for patients with such diseases and strengthening professional education in order to develop nurses specializing in them. This study also recommends that a support system for work and childrearing be developed.

## 1. Introduction

In 2020, the world faced the crisis created by the novel coronavirus (COVID-19) pandemic. By early March 2021, the global number of confirmed COVID-19 infections totaled approximately 117 million, with approximately 2.6 million deaths, and the number of cases continues to increase [1]. In Korea, approximately 94,000 individuals have been infected and approximately 1600 deaths have occurred since the first case was reported in December 2019 [2]. The World Health Organization (WHO) declared the COVID-19 outbreak a public health emergency of international concern (PHEIC) on 30 January 2020, and then characterized it as a pandemic a few months later, on 11 March [3]. Airborne infectious diseases such as COVID-19 have become a global health challenge and a threat to healthcare professionals, causing breakdowns in healthcare systems that have required changes in hospital operations [4].

Not only are the frontline healthcare professionals who care for COVID-19 patients experiencing anxiety and depression at clinical levels [3,4,5,6], but they are also facing ethical dilemmas [5,7] and are exposed to serious risks of developing other mental health problems [5]. During previous infectious disease outbreaks, such as severe acute respiratory syndrome (SARS) [8], Middle East respiratory syndrome coronavirus (MERS-CoV) [9,10,11], and the Ebola virus disease (EVD) [12], nurses reported loneliness, anxiety, fear, fatigue, sleep disturbance, and other physical and mental health issues, as well as tremendous psychological difficulties [5,13]. In general, nurses caring for patients with a new infectious disease should receive pre-work training, adaptive training, and training in the negative pressure ward [6,10,14]. Considering that such education processes can take considerable time, nurses caring for COVID-19 patients should continue to do so until the COVID-19 pandemic ends [5,15].

However, COVID-19 is different from previous infectious diseases. The COVID-19 pandemic has created worsening shortages of nursing staff, which have become increasingly serious [4]. Nurses who care for patients with novel infectious diseases have opportunities to increase their nursing knowledge and improve their practices [8,9,10,11,12,13]; however, they also experience exhaustion [9,16]. Nurse burnout not only reduces the quality of healthcare and causes patient safety problems, but it also affects turnover [17]. Human beings are adaptive and continue to grow and develop in changing environments. Adaptation consists of growth and maturity as well as a positive response that promotes human and environmental changes [18]. During the COVID-19 pandemic, nursing intervention is required to help nurses interact positively with their environment and to promote positive adaptation to health and psychological stress situations. This requires approach strategies that are based on an understanding of the care adaptation process among nurses who care for COVID-19 patients. These strategies will help prevent the turnover of skilled nurses; developing intervention programs can also help them adapt even if new nursing staff are in place. It is thus necessary to understand the adaptation processes nurses go through while caring for COVID-19 patients, as this could improve the quality of nursing services and alleviate nursing staff shortages.

Understanding the adaptation processes of nurses who are caring for COVID-19 patients requires us to examine how they interact with the caring experience. This will allow us to define the important conditions occurring in their adaptation processes and determine how nurses adapt over time [19].

Grounded theory is a methodology used to identify the problems shared among group members experiencing a specific phenomenon and to explore the psychosocial process in depth. It is based on symbolic interaction theory, which posits that human behavior is determined through interactions with others [20]. As other persistent pandemics like COVID-19 are anticipated, a grounded theory approach to nurses’ adaptation processes in caring for patients with COVID-19 is crucial for developing intervention strategies that can help them cope with emerging infectious diseases. This approach will help us explore their psychosocial adaptation processes [19,20]. However, previous studies on the experience of nurses’ adaptation processes in caring for COVID-19 patients [6,14] do little to provide a collective understanding of how nurses, in this unique context, interact with the caring experience and their psychosocial adaptation processes.

Accordingly, this study aimed to develop a substantive theory and obtain an understanding of nurses’ adaptation processes in caring for COVID-19 patients by examining how these nurses cope and adapt using a grounded theory approach. We hope to provide the basic data needed to develop intervention programs that can support nurses who are caring for COVID-19 patients.

## 2. Materials and Methods

### 2.1. Design

This qualitative study explored the adaptation processes of nurses engaged in caring for COVID-19 patients by applying the grounded theory approach proposed by Corbin and Strauss [20].

### 2.2. Participants

The study’s participant selection criteria were as follows. Participants had to have experience working as a nurse at a healthcare institution for at least one year and caring for COVID-19 patients at a nationally designated COVID-19 hospital in the negative pressure ward for at least six months; they also had to be informed about the study’s purposes and procedures and give voluntary consent to participation in the study. Study participants were selected using a theoretical sampling technique. Theoretical sampling is a data collection procedure used to develop concepts and themes from data, in which locations, persons, and events are sampled to identify the relationships among the concepts, confirm diversity, and maximize the chances of developing concepts in terms of properties and dimensions [16]. In this study, sampling continued until theoretical saturation was reached, at which point no new concepts were being extracted via the interviews. A total of 23 nurses participated in the study. First, interviews were conducted with the nurses who met the participation selection criteria among those introduced by the head nurses from negative pressure wards in nationally designated COVID-19 hospitals or through the researchers’ respective professional networks. Later interviews were conducted by sampling participants, such that the study sample comprised nurses with diverse backgrounds in terms of factors such as nursing experience, job rank, and age.

The 23 participants were all female. Twelve (52.2%) participants were in their 30s, the largest age group, and the mean age was 32.34 ± 5.16. The largest share of nurses had three to five years (*n* = 8, 34.8%) of nursing experience; this group was followed by those with six to nine years (*n* = 7, 30.4%) and 10 to 19 years (*n* = 6, 26.1%). The mean years of nursing experience was 8.91 ± 5.43. Fifteen (65.2%) participants were unmarried, and four (17.4%) were married and lived with their children. Twenty (87.0%) participants were staff nurses, and the remainder were charge nurses. Regarding education level, 21 (91.3%) had a bachelor’s degree, and two (8.7%) had a master’s degree.

### 2.3. Recruitment

Data were collected through individual face-to-face in-depth interviews between 7 August and 31 October 2020. The interviews occurred at a place, date, and time chosen by each participant and lasted from 1 to 1.5 h. If further clarification was needed after an interview, an additional interview was conducted over the phone. An isolated area at the participants’ preferred location was used so that they could feel comfortable and speak freely during the interview. All interviews were conducted by the researchers. The main interview question was “Please tell me about your experiences of caring for COVID-19 patients”. While listening to the answers to the main question, the interviewers followed up with exploratory questions (such as “What happened next?” “What did you feel then?” and “What did you think?”) appropriate for the situation. To create an atmosphere in which participants could talk comfortably, each interviewer started with small talk and began the interview only when the participant appeared to be comfortable. During the interview, the interviewer avoided being critical or interpretive of the participant’s statements to preclude bias caused by the interviewer’s preconceptions. Additionally, the interviewers made detailed field notes on participants’ behaviors, facial expressions, tones, and other factors. After each interview was over, the interviewers wrote memos about anything they considered notable and their feelings during the interview; these notes were used as supplementary data during the analysis. All interviews were audio-recorded with the participants’ consent, and the recordings were immediately transcribed by two research assistants. The researchers confirmed the accuracy of the transcribed data by comparing each transcription with the associated audio recording in its entirety. Data collection was terminated when it was determined that theoretical saturation had occurred—when no new meaningful data were being observed, and no new categorical properties or dimensions were being identified.

### 2.4. Data Analysis

Following the grounded theory methodology proposed by Corbin and Strauss [20], data analysis was conducted simultaneously with data collection in a cyclical manner, proceeding through the steps of open, axial, and selective coding. During open coding, the researchers identified, labeled, and conceptualized the phenomena experienced by the participants by reading the transcribed data line by line; the researchers then categorized the data in terms of their similarity and dissimilarity. Line-by-line coding reduced the likelihood of superimposing preconceived notions in data analysis. During axial coding, the structure (contextual conditions, causal conditions, central phenomena, intervening conditions, strategies for action and interaction, and results) of the categories derived via open coding was developed using the paradigm model, an auxiliary analytical framework. In addition, the ways in which participants adapted to the phenomena of interest and changed over time were identified. During selective coding, hypothetical statements were compiled in order to comprehensively explain the process of nurses’ adaptation in caring for COVID-19 patients. Finally, the researchers formulated a theory regarding this process by deriving an integrative and highly abstract core category in which to integrate the relationships among all the extracted categories.

### 2.5. Study Validity and Rigor

To ensure the rigor of the study findings, the researchers made efforts to increase their truth value, applicability, consistency, and neutrality, as suggested by Guba and Lincoln [21]. Specifically, to increase truth value, the researchers acquired sufficient background knowledge pertaining to the research question by perusing numerous literature reviews, news reports, and datasets featuring theoretical sensitivity concerning the contexts and meanings that apply in interview situations. In addition, the researchers checked for any errors in the transcribed data within 24 h after each interview ended by repeatedly listening to the audio recording and comparing it with the transcription. They also verified the consistency between the transcriptions and outcomes derived from the analysis by comparing them iteratively. To increase applicability, the participants were selected by considering their work experience so that their adaptation in caring for COVID-19 patients could be described from a variety of perspectives. Moreover, data were collected until participants’ statements about their experiences reached a point of theoretical saturation. To increase applicability even further, the analysis outcomes were shown to two nurses with clinical experience of 15 years or more who did not participate in the study but met the participant selection criteria to confirm that the analysis outcomes were meaningful in light of their own experiences. To ensure consistency, the researchers continued contemplating the main research question during the entire research process and conducted data collection and analysis in cycles. In addition, they held regular meetings to continuously discuss the data collection and analysis processes. Finally, to maintain neutrality, the researchers made efforts to preclude personal biases relevant to the study topic by recording their preconceptions, assumptions, and biases in a research diary throughout the research process.

### 2.6. Ethical Considerations

The study was approved by the institutional review board (IRB) of the institution with which the researchers are affiliated (IRB No. 1041478-2020-HR-025). Prior to conducting the interviews, the interviewers informed the participants about the study’s purposes and procedures and said that the interviews would be audio-recorded. The participants signed a written consent form and were informed that the interview content would be used for research purposes only and anonymized, and that they could terminate their interview at any time. They were also informed that all data would be saved in a password-protected computer designated by the researchers and that they would not be used for any purpose other than for research. The participants were further informed that the data would be discarded after the study findings were reported. The participants were given a small gift as an expression of appreciation.

## 3. Results

Categorization of the interview data through an open coding process yielded 47 subcategories and 13 higher-level categories (see Table 1). The relationships between categories were established through axis coding according to the paradigm model, and a context model was also constructed by combining the categories (see Figure 1). “Growing as a proficient nurse alongside comrades on the COVID-19 frontline” was found to be the core category in nurses’ adaptation while caring for COVID-19 patients. The adaptation process was found to comprise periods of “confusion”, “burnout”, “leaping forward”, and “stabilization”.

### 3.1. Causal Conditions

A causal condition is one that induces a particular phenomenon [20]. In this study “the COVID-19 frontline, toward which nurses were suddenly pushed” was the causal condition inducing the central phenomenon in nurses’ adaptation to caring for COVID-19 patients. This causal condition forced the participants to face the situation without any preparation or choice in the matter; hence, they experienced a sudden, dramatic change:
I didn’t want to care for COVID-19 patients and didn’t know how to care for COVID-19 patients…When severe patients were suddenly assigned to me in the ward, I had to make a phone call to another ward to borrow a CRRT (continuous renal replacement therapy) machine. I received an order to perform ventilator therapy when I was not even ready to prepare for suction, and honestly, I had no idea what to do. (Participant 22)

### 3.2. Central Phenomena

Central phenomena comprise participants’ behaviors, statements, experiences, or repeated patterns in response to a problem or situation that occurred due to the impact of the causal condition [20]. The central phenomena in nurses’ adaptation process when caring for COVID-19 patients were “complexed emotions”. The subcategories of “complexed emotions” were “anxiety over the lack of medical supplies and equipment”, “fear of something that has not been experienced before”, “the burden of new tasks”, “frustration with the endless situation”, “anxiety over unpredictable patient prognoses”, and “feelings of shame, faced with their own limitations”. At the beginning of the pandemic, nurses were not provided with enough personal protective equipment (PPE), such as masks and protective gowns. Additionally, as patient conditions worsened, mid-career nurses with little experience in caring for severe patients essentially became novices, as they were attempting to use unfamiliar machines. It was difficult for them to determine and predict the status of patients with whom they had no previous experience. COVID-19 patients have diverse symptoms, and the guidelines for symptomatic therapy are unclear and can be confusing. As experienced nurses suddenly became novices, they felt ashamed and helpless, viewing all the experience they had accumulated as useless. Instead of improving, this situation persisted:
Because we are not ICU (intensive care unit) nurses, we didn’t know how to prepare when we heard that a CV (central venous) catheter should be inserted. We didn’t even know what Prolene and silk were. I am a charge nurse and yet I did not know how to cope with such a situation, like acting nurses. I didn’t know how to keep a nursing record in the current situation and I didn’t even know what nursing instructions I should give. It was very hard and so I cried a few times during work hours. Every day I was afraid of work. (Participant 18)

### 3.3. Contextual Condition

A contextual condition is a collection of concrete temporal relationships or situations influencing the central phenomenon or issue [20]. In this study, “difficulties in the workplace” and “disruption in daily life” were extracted as the contextual conditions pertaining to the central phenomena. The worse the contextual conditions, the worse the experience of the central phenomena.
Difficulties in the workplace

The subcategories of “difficulties in the workplace” were “poor work environment”, “uncertainty of manual and guidelines”, “patient’s severity”, and “insufficient and unfair compensation”. Although the participants received regular training, they were not used to performing bedside nursing activities while wearing protective gear. Thus, it was cumbersome for them to put on and remove PPE. Further, their goggles would become foggy, obstructing their vision if worn for long periods. After performing nursing care in PPE for a long time, participants experienced hypoxia, dizziness, and difficulty breathing. If they gave an intravascular injection or performed other delicate nursing activities while wearing multiple layers of gloves, they had to work much slower than usual, expending great amounts of physical energy. There was no guarantee of days off, and the work schedule could be revised without notice, which meant that they had to be on constant standby for emergency calls. Additionally, because the nurses worked near patients in PPE, they often had to perform duties ordinarily performed by physicians, laboratory technicians, nurses’ assistants, and cleaners:
Working with goggles, a mask, and a face shield on, I sweat so much. Sweat gets in my eyes and the visual fields through the goggle lenses are whitish. In that condition, I have to change patients’ diapers and positions. If it goes over the maximum of 2 h, I feel dizzy and have a hard time breathing. (Participant 11)

The Korea Centers for Disease Control and Prevention (KCDC) released infection control guidelines, but they do not include all the details relevant to the COVID-19 pandemic and thus could not be directly applied to bedside care, which caused additional confusion. Generally, each healthcare institution has its own guidelines for infection control. However, those guidelines did not provide guidance for all possible scenarios, adding to the confusion. Furthermore, the guidance for patient nursing practices changed frequently, and new hospital guidelines for infection control were sometimes communicated to the nurses incorrectly:
We have the KCDC guidelines for infection control, but the guidelines do not contain details. So, ultimately, the definitions for the concepts of “contamination” and “clean” are determined by each hospital in accordance with their situation, and we act accordingly. So, we were very confused on top of being busy at work. In addition, every day, I go to work and find that the job manual has changed. For example, the instructions for doing certain nursing tasks for inpatients would change on a daily basis. (Participant 3)

The nurses experienced extreme fear and frustration when a COVID-19 patient’s condition worsened. When a patient died suddenly, the nurses experienced increased psychological suffering, which increased their feelings of guilt.

The government announced compensation programs for COVID-19 healthcare professionals. However, the programs did not adequately reflect the level of difficulty in healthcare practice, and compensation was paid either to those at screening centers or only to healthcare professionals who were dispatched with the support of a city government or the central government. The programs created a feeling of relative deprivation among nurses working as COVID-19 healthcare professionals in hospitals. In addition, many of the supplemental nurses who had been hired with governmental support were returning to bedside nursing after taking a break from their nursing career, so they required infection control training, increasing the nurses’ job load rather than helping alleviate it. The nurses required in practice were those with professional knowledge and skills who could care for severe patients. The lack of a workforce with these skills was a serious problem:
The bi-weekly wage of those dispatched by the city government was almost the same as what we get by working for a month. They ask for too much dedication from us. We don’t even get a risk payment. (Participant 8)
What we needed so urgently was not a high number of nurses working unconditionally. We needed nurses with ICU experience. But the supplemental nurses were those on a break from their nursing career who could not handle even IV lines well. Ultimately, our job load remained the same. (Participant 17)
Just because we had to wear level D and PPE, I sometimes did physicians’ jobs like Ambu-bag bagging and arterial puncture for them. If an accident happens while we do their jobs, however, who will protect us? And yet, we had to do as we were told. (Participant 9)
Disruption in daily life

The subcategories of “disruption in daily life” were “isolation from family and friends”, “uncomfortable stares from people around them”, and “meticulously observing social distancing”. As healthcare professionals, the participants practiced social distancing in their daily lives more faithfully than anyone else in order to protect patients. In addition, some participants who had family members with a chronic illness moved out of their homes to stay away from them. Furthermore, participants stopped meeting friends in person and limited their activities to going back and forth between the hospital and their accommodations. Having restricted outdoor activities involving large gatherings, they stopped participating in hobbies that had previously helped them destress. Affected by these contextual conditions, nurses experienced the central phenomena:
I could not go to my house. I got a place for myself and lived away from my family. My grandmother who cares for my children is also at high risk, vulnerable to infection, and my children are still young. So, my family told me not to come home. Since my friends, whom I often met, found out that I care for COVID-19 patients, we contacted one another only through SNS (social network service). (Participant 15)

### 3.4. Intervening Condition

An intervening condition is one that deintensifies or induces a change in a phenomenon being experienced; it is a factor that facilitates or inhibits action/interaction [20]. The intervening conditions for nurses’ adaptation process in caring for COVID-19 patients in this study were “ability to cope”, “camaraderie”, and “support system”.
Ability to cope

The subcategories of “ability to cope” were “application of past experiences of caring for severe patients and MERS patients”, “wits in an emergency situation”, and “learning how to care for severe patients”. Nurses who had experience handling ventilators, CRRT, and ECHMO (extracorporeal membrane oxygenation) adapted to the new tasks faster than the others. Experience providing bedside care in PPE during the MERS outbreak was of great help in the new situation. Even amidst an emergency situation that no one had experienced before, proficient nurses were able to cope very quickly. Managing the anxieties of sensitive patients infected by COVID-19 was a burdensome job for nurses, but those patients were no different from difficult patients in general for experienced nurses who had seen a wide variety of patients:
In the beginning, it was tough to care for COVID-19 patients because I didn’t know how to care for them, but my past 20 years of experience could not be ignored. It was always my role to control the COVID-19 patients who were really fussy. (Participant 16)
Camaraderie

The subcategories of “camaraderie” were “colleagues whom I can rely on for my safety”, “colleagues who shared tough times with me”, and “colleagues who complete their share of work”. Participants sustained themselves through camaraderie, even though they had difficulties adapting to providing care for COVID-19 patients. This was not a feeling they shared simply because they worked together. Participants meticulously followed the infection prevention measures because they believed that an infection in one of their colleagues would put them all at risk. Participants depended on their colleagues and tried to create a safe work environment. Though it was grueling to cope with emergency situations while wearing protective gear, they could sustain themselves with the thought that they were not alone because their colleagues were with them. In addition, they gained the strength to carry on because each of them was performing their given role to the full:
If, in the patient’s room, a COVID-19 patient undergoes bronchoscopy or we prepare for CRRT, we have so many things to do. If I asked a colleague to do a nursing task and had to check whether or not the task was properly done, how could I do my job? I can only do my job well because the nurse competently did the task. (Participant 21)
Support system

The subcategories of “support system” were “active support from nurse managers”, “family members’ interest and understanding”, “the encouragement of the people around them”, and “availability of childcare centers for work and childrearing”. When participants watched the nurse manager echo the guidance offered and revamp and coordinate a variety of issues to improve the job environment, they felt comforted knowing that everyone was hard at work. Further, participants were greatly inspired by the knowledge that their family was sincerely concerned about them and that they understood what they were doing at work. They were also grateful when friends and others recognized their hard work and dedication:
My friends sent gifts and encouraging messages through SNS, thanking us for our sacrifices and dedication. (Participant 13)

Some participants were raising young children and had difficulty making childcare arrangements when they had to go to work. Childcare centers might be closed, and friends or family might not be able to babysit if a close contact of a confirmed case had been identified. Some childcare centers to which participants had previously sent their children refused to accept them because their mother’s profession elevated the child’s risk of infection. In some cases, even participants’ parents refused to help with childcare because they were afraid of infection. Additionally, since schools were no longer holding face-to-face classes, participants were also responsible for their children’s education:
It was really tough to accept that no institution can provide childcare in this situation; meanwhile, they are concerned about the serious problem of the country’s low birth rate. Yet, no one can care for my children. I asked my 7-year-old to babysit my 4-year-old so I could go to work. Although my husband came home in the morning after working a night shift, I was scared that something might happen during the few intervening hours. (Participant 15)

### 3.5. Strategies for Action and Interaction

An action and interaction strategy is a strategy or routine/habitual behavior used to cope with a problematic situation; it describes with whom and how those facing a problem interact in the context of the challenging phenomenon [16]. Participants facing the new challenge of “adapting to caring for COVID-19 patients” used the strategies of “dashing first”, ”enduring on the very last of my strength”, “growing stronger to get back up”, and “dreaming about the end of the COVID-19 pandemic”. In terms of time flow, these strategies were used during nurses’ adaptation processes in caring for COVID-19 patients in four periods: confusion, burnout, leaping forward, and stabilization.
Period of confusion: Dashing first

The confusion period is the stage in which nurses caring for COVID-19 patients went through trial and error when they had no idea what to do with the new infectious disease. During this time, the participants were unable to predict their daily tasks, so they used an interaction strategy called “dashing first”. Each time the nurses went to work, they had to become familiar with the changed work protocols, and they had to quickly perform admission and discharge care for patients in the evening. In this stage, to increase the nurses’ ability to adapt at work, three shifts often suddenly changed to two, and nurses who were on vacation would suddenly need to report to work due to understaffing. Whenever such a situation occurred, it was accepted as an unavoidable problem, and, once it became reality, it became a process of learning to work passively.

The subcategories of “dashing first” were “handling the situation at hand first”, “accepting a life with no break from work”, “always standing by for a sudden call to work”, and “asking other departments for help”. It was not easy for participants to adapt to the daily-changing job manual, but they started focusing on caring for patients. They often exchanged opinions with colleagues after work through group SNS accounts, and were thus constantly performing their jobs at home. As they continued to work overtime, the work schedule was changed to a two-shift pattern to increase efficiency in patient care. However, when that work framework was established and the system had stabilized, it changed again, this time to a three-shift pattern. Nonetheless, there was always a shortage of proficient nurses to care for severe patients. Participants were called into work without notice due to insufficient staffing. They requested that other departments, like the ICU, help by dispatching their personnel because they were unfamiliar with the equipment needed to care for severe patients:
The night shift starts at 10:30. I got a phone call from the ward around 10 to come in and work. That was supposed to be my day off, so I was getting ready to go to bed. But as soon as I got the phone call, I got ready and went to work. This has happened often. (Participant 4)
Period of burnout: Enduring on the very last of my strength

The burnout period is a stage that appears over time after a period of confusion has been passed through. At this stage, nurses used the interaction strategy of “enduring on the very last of my strength”. The subcategories of “enduring on the very last of my strength” were “repetition of work and sleep” and “waiting to take days off”. In the burnout period, the nurses were so physically and psychologically exhausted that they had no energy to do anything. They endured the time by repeating a pattern of sleeping immediately after work and going to work immediately after waking, while having no time for hobbies or a personal life; when they worked, they waited only for days off. Some nurses made a more rapid transition from burnout to leaping forward than others depending on their coping ability, camaraderie, and support system.
Period of leaping forward: Growing stronger to get back up

The period of leaping forward is a step towards using aggressive coping strategies in clinical conditions, wherein participants showed an interaction strategy of “growing stronger to get back up”. In this stage, nurses gained specialized nursing knowledge about new infection management and critical care, or sought effective nursing practice strategies by reflecting on emergency situations. In addition, the participants not only promoted their inner strength at this stage but also encouraged and supported their colleagues, recharging their power to rise together.

The subcategories of “growing stronger to get back up” were “accepting the self as imperfect”, “acquiring professional nursing knowledge”, “reflecting on the situation”, “blocking out unnecessary sources of emotion”, “recharging energy”, “sympathizing with colleagues”, and “considering colleagues’ perspectives first”. Although participants were fatigued as they experienced trials and errors in the persistent COVID-19 situation, they used various interaction strategies to grow stronger and continue their work. They let go of their self-image as the proficient nurses they had been in the past and accepted that they were now novices who were unfamiliar with the nursing care required by COVID-19 patients. They began addressing their shortcomings. To acquire the professional nursing knowledge needed to care for severe patients, they participated in hospital training for intensive care nursing. After experiencing an emergency situation, they re-examined the situation and contemplated problem-solving approaches. To minimize their emotions while caring for COVID-19 patients, they tried to view problems objectively. Since they were physically exhausted by the time they were used to the job, they began to exercise again to recharge their energy stores. Further, in sharing their frustrations and talking with colleagues, participants enjoyed being recognized for their efforts to do their best in a demanding situation and were able to comfort themselves. They performed small, considerate acts for their colleagues, such as offering a glass of water to a colleague who was tired after wearing PPE for more than two hours and recording a video demonstrating how to operate a particular ICU machine for the benefit of colleagues who were not yet proficient:
I was so exhausted that I wanted to give up everything. But I improved my physical strength by exercising. Because I could not go to any place where there were many people, to protect myself from getting infected with COVID-19, I began to do exercises I could do alone. I utilized home training as much as I could by watching YouTube videos. (Participant 19)
Period of stabilization: Dreaming about the end of the COVID-19 pandemic

The stabilization period occurred when the nurses were able to proficiently perform nursing tasks with which they were unfamiliar at the beginning of the pandemic. This stage involved skillfully performing new infection management and critical care services with which they were not familiar when they began nursing patients with COVID-19. In this stage, the nurses showed an interactive strategy of “dreaming about the end of the COVID-19 pandemic” in a stable way, working with little hope that this situation would ever end. The nurses comforted each other by talking about dreams they had not yet realized due to COVID-19 and about how the pandemic would end once a vaccine had created a collective immunity.

The subcategories of “dreaming about the end of the COVID-19 pandemic” were “holding on to hope”, “imagining going on a trip”, and “making a bucket list”. Although the end of COVID-19 was not yet in sight, participants withstood daily life by not losing hope that the pandemic would end. Participants said that, had they not held onto even the vague hope that the pandemic would end, they would not have had the strength to live each day. Many participants reminisced about the trips they had taken and spoke about their bucket lists of travel destinations they planned to visit when the situation improved. Participants dreamt about the future by creating other bucket lists, stating that they remembered the things they could not do because they were busy caring for COVID-19 patients:
Won’t the COVID-19 situation be over some time down the road? Without hope, how can we live today? (Participant 16)

### 3.6. Consequences

The results derived through action/inaction were “growth as an expert nurse” and “changing career path”.
Growth as an expert nurse

The subcategories of “growth as an expert nurse” were “job stabilization”, “establishment of a viewpoint on nursing professionalism”, and “heightened awareness of ethical responsibility”. Over time, participants felt that they were growing, as they systematized and familiarized the tasks involved in caring for COVID-19 patients. They found the meaning of nursing in the familiar work, recognized the professionalism of nursing, and took pride in being nurses. In addition, participants pledged that, as nurses caring for patients on the frontline of the COVID-19 pandemic, they would make sure to fulfill their ethical responsibility:
Now, I am familiar with the job to some extent, due to the manual we created. I am also used to wearing level D and PPE. When I receive a CRRT or ECHMO order, I don’t get scared; instead, I can handle it with proficiency. (Participant 1)
In this situation, I think our job of watching over patients is really meaningful. I also believe that our job has professionalism. As I overcame the uphill battle, I realized that many healthcare professionals rely on nurses’ professionalism. (Participant 7)
Changing career path

The subcategories of “changing career path” were “a decision to change jobs” and “preparing for another job.” Some nurses decided to leave bedside nursing although they had become familiar with the nursing tasks. Reflecting on the past, during which they tirelessly worked caring for patients, they decided to leave a situation in which they were pressured, without any compensation, to make sacrifices and dedicate themselves to a national disaster only because they were nurses. They wanted to make a change in their lives. Some nurses tried to find another job and studied for another field:
I am going to quit working at the hospital once the pandemic is over. I want to have time to look after myself. There is talk about compensation at the government level, but I don’t expect anything. I sacrificed my life to care for patients, but we got nothing back. (Participant 3)

## 4. Discussion

This study was conducted to explore how nurses adapted in order to care for COVID-19 patients during the COVID-19 pandemic using a grounded theory approach. The aim was to develop a substantive theory and provide the basic data needed to develop intervention programs to support nurses caring for COVID-19 patients.

The study found that “growing as a proficient nurse alongside comrades on the COVID-19 frontline” was the core category. A previous study that analyzed nurses’ experience of caring for COVID-19 patients using a phenomenological methodology [15] also found that the essential structure of the phenomenon was growth after the frontline battle against an infectious disease pandemic. This experience was also identified in nurses who cared for MERS patients during the MERS outbreak [10,11,22]. Our findings indicate that education, investments, and job environments for nurses have not improved since the MERS outbreak. It is anticipated that new infectious diseases will continue to emerge [23]. To prepare for emerging infectious disease outbreaks, nurses’ job environments should be enhanced, and greater investments should be made. Additionally, it is necessary to reinforce basic nursing care training for emerging infectious diseases as well as professional training for nurses who are specialized in infectious diseases.

The study found that the causal condition was “the COVID-19 frontline toward which nurses were suddenly pushed.” A phenomenological study that examined the experiences of nurses caring for COVID-19 patients [24] similarly found that the core category was “pushed onto the battlefield without any preparation.” Our study’s core category is thus also in line with the experiences of nurses working amid the MERS outbreak, who, though wanting to avoid caring for MERS patients because they feared infection, accepted the new challenge as part of the responsibility of being nurses because it was hard to refuse meeting the hospitals’ needs [22]. This study found that participants were unprepared when they began caring for severe COVID-19 patients; this ill-preparedness increased their psychological stress and fear. Previous studies [7,24,25,26,27] have pointed out that, because such fear induces post-traumatic stress disorder (PTSD), healthcare professionals must be provided with interventions for psychological problems beginning at an early stage of disasters like pandemics caused by emerging infectious diseases. However, governments and the medical field have failed to offer psychological intervention programs to healthcare professionals amidst the COVID-19 pandemic. Such programs should be developed and administered, and a variety of practical ways to support nurses experiencing psychological difficulties should be explored.

The study found that “complex emotions” constituted the central phenomena. Such phenomena were also found in nurses caring for MERS patients [10,11,22]. Similar central phenomena were also found in a study on the experience of caring for COVID-19 patients [28]. Although the study participants received education on PPE use in the context of caring for general infectious disease patients, they were unprepared to care for severely ill patients. Then, they were suddenly tasked with the care of severe COVID-19 patients in the general ward. This sudden change made increased psychological stress likely, due not only to the burden of caring for patients with an emerging infectious disease but also to the fear involved in caring for severely ill patients [14,29]. Psychological stress contributes to burnout among nurses [30], ultimately lowering nursing quality [10,11,22]. Moreover, extremely stressful situations amidst the persistent COVID-19 pandemic threatens nurses’ mental health [6,15,31]. Since severe stress due to physical and mental burnout can lead to PTSD, psychological support in the form of early counseling and intervention is needed [16,32,33]. Considering the current circumstances, wherein providing infection-prevention training in a group setting is difficult, it is necessary to develop web- or smartphone-based programs geared toward mental and physical stability, stress reduction, and burnout reduction on the job [34,35].

The contextual conditions in the study were “difficulties in the workplace” and “disruption in daily life.” These problems were also identified during the MERS outbreak [10,11,22] and have reoccurred during the COVID-19 pandemic. The government communicated infection management guidelines, but detailed and systematic guidelines for the job were lacking. Accordingly, it is necessary to establish a web-based central information system that would allow practitioners to share information regarding patient safety, detailed infection management, and other healthcare-based coping experiences during the pandemic [36]. Improvements should also be made to poor work environments, which feature inadequate staffing, vague boundaries between nursing tasks, low nurse salaries, and other problems; specific plans should be formulated at the national level to address those issues [37]. If nurses continue to be pushed to make sacrifices without appropriate compensation, the nursing shortage will worsen [38].

While adapting in order to care for COVID-19 patients, participants went through periods of “confusion”, “burnout”, “leaping forward”, and “stabilization”, and used “dashing in headfirst”, “enduring on the very last of my strength”, “growing stronger to get back up”, and “dreaming about the end of the COVID-19 pandemic” as interaction strategies. A previous study on workplace learning among nurses reported a similar adaptation process [39]. Nurses undergo a learning process in the field that entails learning through mentoring and being mentored by colleagues. Hence, to promote nurses’ professional growth, professional camaraderie should be encouraged, and highly functional workplace teams should be developed and utilized [39].

The study’s intervening conditions were “ability to cope”, “camaraderie”, and “support system”. If these intervening conditions are insufficient, nurses will remain in the confusion and burnout periods for a long time. Nurses can move through the adaptation process stages quickly if their ability to cope, camaraderie, and support system levels are high enough. The participants’ past experiences of caring for emergency and severely ill patients over the years were integrated into their coping abilities, impacting their adaptation to the COVID-19 situation. When nurses face a new situation, their patient nursing experiences are integrated with their training, triggering professional growth [15]. To facilitate the integration of training and patient nursing experience and bolster nurses’ ability to cope, senior nurses should mentor junior nurses, focusing on helping them with communication skills, encouraging proactive wellness and mindfulness strategies, and helping them deal with the challenging stressors they are facing [39].

Consistent with the intervening conditions derived in this study, a previous study identified “camaraderie”, “patience”, and “encouragement” as factors that helped nurses endure the MERS outbreak, which was a similarly challenging situation [22]. Furthermore, a previous study on the experiences of COVID-19 nurses found that camaraderie and support and inspiration from family, friends, patients, and the public were important factors [14].

However, previous research did not find that the lack of childcare centers and childrearing resources was a support system problem because the MERS outbreak lasted for barely a year, while the COVID-19 pandemic began in March 2020 and was still ongoing in March 2021. The situation examined in this study was more challenging for participants, who had to work while caring for their children because childcare centers had closed. Previous studies found that stress levels amidst the COVID-19 pandemic were higher among married nurses than unmarried nurses due to the former’s increased fatigue and feelings of guilt while attempting to balance work and childcare [24,31]. The Korean government has instituted a variety of work-family support programs as part of its effort to counteract the nation’s low birth rate, but no childcare centers are open 24 h to accommodate people working in a three-shift pattern, and many hospitals do not offer childcare onsite [40,41]. Therefore, ways of providing nurses with the practical support they need to balance work and childrearing should be explored at the national and hospital levels [41,42]. Helping nurses with their adaptation to the COVID-19 situation requires that currently available support systems be fully utilized and that new support systems that could assist them in effectively implementing various coping strategies be established.

Similar to the experiences of nurses caring for MERS patients [10,11,22], the participants in this study grew professionally into expert nurses and felt a heightened awareness of their ethical responsibility as a result of undergoing the adaptation process. According to Benner’s [43] Dreyfus model of skill acquisition, a proficient nurse returns to the novice stage when performing a new job or entering a new field and then gradually grows into a proficient nurse again. The participants in this study were either in the proficient or expert stage of Benner’s [43] professional socialization process; however, when thrust into a new situation in which they did not know what to do, they had to learn new nursing tasks through trial and error, like novices, before becoming proficient. Thus, support should be provided at the organizational level to facilitate nurses’ steady growth. Given what is required in the COVID-19 situation, nurses’ professional growth should be supported through the provision of web-based educational programs designed to strengthen nursing competence.

The COVID-19 pandemic has lasted longer than the MERS outbreak, and some of the nurses examined in this study decided to leave bedside nursing or were preparing to change jobs. Given the severe nursing shortage, measures to support nurses who plan to leave the bedside or change jobs are urgently needed. A previous study on nurse turnover [38] reported that, despite the implementation of coping strategies such as competence and self-control, nurses were disappointed about promotions in clinical care and eventually decided to leave bedside care. One nurse’s turnover not only places a financial burden on the healthcare institution, but it also affects other nurses’ turnover intentions [38]. A policy approach is required for nurses with turnover intention who are physically and mentally burned out due to the lack of adequate downtime amidst the COVID-19 pandemic. The Infectious Disease Control and Prevention Act of Korea offers no specifics pertaining to how to compensate and support healthcare professionals working in patient screening and treatment during the outbreak of an infectious disease like COVID-19. Specific ways to compensate nurses who make sacrifices during global crises, such as the COVID-19 pandemic, should be formulated at the national and social levels.

This study has several limitations. First, in Korea, over 20,000 men have been licensed as nurses, comprising approximately 5% of the country’s nursing profession [44], but the participants in this study were all female. In Korea, most male nurses work in a special ward, such as the emergency room (ER), ICU, or surgery unit; thus, male nurses were not included in this study because its subjects were nurses who had cared for COVID-19 patients in the general ward. Hence, this study did not consider between-sex differences in adaptation; these should be explored in a future study. Second, the study’s findings were focused on the adaptation process of nurses working in a ward in a regional tertiary hospital designated by the government to treat COVID-19 patients only. Thus, caution should be exercised in interpreting the data. Future research should examine the adaptation process of nurses caring for COVID-19 patients in different regions, the differences between adaptation processes depending on hospital size, and the adaptation processes of nurses working in a special ward such as the ER or ICU. Third, the data used in this study were collected eight months after the first confirmed COVID-19 case in Korea but did not cover the third outbreak of COVID-19. Accordingly, any interpretation of the results should consider that the data do not cover the third wave of the pandemic. Lastly, as this study used a qualitative research method, it may have analysis and interpretation biases.

## 5. Conclusions

This study was conducted to understand nurses’ adaptation processes in caring for COVID-19 patients. A grounded theory approach was taken in order to provide the basic data needed to develop intervention programs designed to support nurses caring for COVID-19 patients.

The results of this study show that nurses transitioned from the burnout to the leaping forward stages more rapidly if they had greater coping ability, camaraderie, and support system levels. Therefore, supporting nurses in their systematic coping with emerging infectious diseases requires the regular provision of basic education regarding the care for patients with these diseases and the creation of educational programs that can train nurses specialized in the field of infectious disease. This study also recommends creating a support system for both work and childrearing, the importance of which is shown in the intervening conditions. The evidence offered through this study’s investigation of nurses’ adaptation processes in caring for COVID-19 patients can assist in developing strategies for helping nurses adapt in their jobs and in preventing nurse burnout when another emerging infectious disease occurs. The study’s findings can also be used to establish programs that can prevent the turnover of expert nurses with experience in caring for COVID-19 patients.

## Figures and Tables

**Figure 1 ijerph-18-10141-f001:**
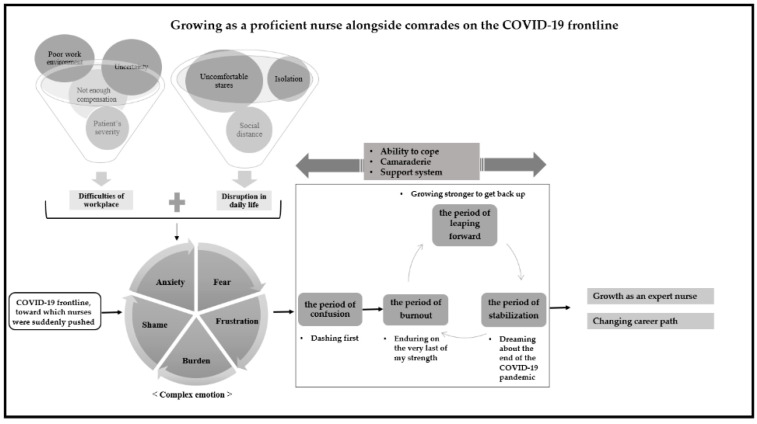
Nurses’ adaptation process in caring for COVID-19 patients.

**Table 1 ijerph-18-10141-t001:** Paradigm, categories, and subcategories of nurses’ adaptation processes in caring for COVID-19 patients.

Paradigm Element	Category	Subcategory
Causal condition	Nurses suddenly pushed towards the COVID-19 frontline	Situation without any preparation
		Situation without choice
		Everything changed suddenly
Central phenomena	Complex emotions	Anxiety over the lack of medical supplies and equipment
		Fear of something that has not been experienced before
		The burden of new tasks
		Frustration with the endless situation
		Anxiety over unpredictable patient prognoses
		Feelings of shame, faced with their own limitations
Contextual conditions	Difficulties in the workplace	Poor work environment (e.g., work schedule with no breaks, PPE, etc.)
		Uncertainty of manual and guidelines
		Patient’s severity
		Inadequate and unfair compensation
	Disruption in daily life	Isolation from family and friends
		Uncomfortable stares from people around them
		Meticulously observing social distancing
Intervening conditions	Ability to cope	Application of past experiences of caring for severe patients
		Wits in an emergency situation
		Learning how to work with difficult patients
	Camaraderie	Colleagues whom I can rely on for my safety
		Colleagues who share tough times with me
		Colleagues who complete their share of work
	Support system	Active support from nurse managers
		Family members’ interest and understanding
		Encouragement from the people around them
		Childcare centers’ help with work and childrearing in parallel
Strategies for action and interaction	Dashing first	Handling the situation at hand first
		Accepting a life with no break from work
		Always standing by for a sudden call to work
		Asking other departments for help
	Enduring on the very last of my strength	Repetition of work and sleep
		Waiting to take days off
	Growing stronger to get back up	Accepting the self as imperfect
		Acquiring professional nursing knowledge
		Reflecting on the situation
		Blocking out unnecessary sources of emotion
		Recharging energy
		Sympathizing with one another
		Considering colleagues’ perspectives first
	Dreaming about the end of the COVID-19 pandemic	Holding on to hope
		Imagining going on a trip
		Making a bucket list
Consequences	Growth as an expert nurse	Job stabilization
		Establishment of a viewpoint on nursing professionalism
		Heightened awareness of ethical responsibility
	Changing career path	Decision to change jobs
		Preparation for another job

## Data Availability

The data presented in this study are not publicly available for privacy reasons.

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
