# Peer review of "Nurses’ Adaptations in Caring for COVID-19 Patients: A Grounded Theory Study"

_ijerph, 2021, doi:10.3390/ijerph181910141_

Round 1

Reviewer 1 Report

The study is of great interest to the scientific community today and is rigorously designed and written. Only recommend to the authors that they have a greater impact on the future open and necessary lines of research, as well as on the usefulness of the results found in the health teams.

Author Response

Dear Reviewer,

We would like to thank you for your time and constructive feedback in improving our manuscript, “Nurses’ Adaptation in Caring for COVID-19 Patients: A Grounded Theory Study” (ijerph- 1370438), which we submitted to the International Journal of Environmental Research and Public Health. We will remember your suggestion and try to conduct research more responsibly.

Please do let us know if there is anything else we can do at this stage. We would be more than happy to do everything we can to assist you in this process.

Best Regards,

Suhyun Kim

Reviewer 2 Report

Authors added a very important paragraph on the limitations of the study. With this included, l am happy to recommend publication of manuscript. 

Author Response

Dear Reviewer,

We would like to thank you for your time and constructive feedback in improving our manuscript, “Nurses’ Adaptation in Caring for COVID-19 Patients: A Grounded Theory Study” (ijerph- 1370438), which we submitted to the International Journal of Environmental Research and Public Health. We're happy to see the review comments that we've met your suggestions, too.

Please do let us know if there is anything else we can do at this stage. We would be more than happy to do everything we can to assist you in this process.

Best Regards,

Suhyun Kim

Reviewer 3 Report

Defeating the SARS-CoV2 virus and ending the COVID-19 pandemic is the most important global challenge that the entire global healthcare system has been facing for over a year. The next mutations or complications that healers struggle with are the problems of today's medicine. At the same time, the medical staff is constantly trying to help other patients and ensure that they have access to the necessary therapies. However, the dynamic development of the pandemic situation, staff shortages and uncertainty about the health consequences - these are factors that cause severe stress. Those who deal with it experience burnout more often and require psychological support. The role of an efficient system is to provide them with such care. Therefore, it is necessary to monitor the physical and mental condition and problems faced by people on the front lines, and manuscripts such as the present one are an important source of knowledge about these phenomena.

The manuscript was prepared in a transparent manner, with a typical layout for experimental work, while maintaining the correct proportions of the individual parts. The aim of the study was formulated correctly, detailed goals were described and a hypothesis was formulated. The aim has a logical justification and is based on the theses of the work. I assess the topic of the work as extremely topical and important due to the scale of the problem. The research methods were selected properly for the intended purpose. The set research goal was achieved. Correctly formulated conclusions can be drawn from the obtained data, which has been clearly presented and thoroughly discussed. The assessed work fully meets the requirements for this type of article.

Author Response

Dear Reviewer,

We would like to thank you for your time and constructive feedback in improving our manuscript, “Nurses’ Adaptation in Caring for COVID-19 Patients: A Grounded Theory Study” (ijerph- 1370438), which we submitted to the International Journal of Environmental Research and Public Health. We're happy to see the review comments that we've met your suggestions, too.

Please do let us know if there is anything else we can do at this stage. We would be more than happy to do everything we can to assist you in this process.

Best Regards,

Suhyun Kim

This manuscript is a resubmission of an earlier submission. The following is a list of the peer review reports and author responses from that submission.

Round 1

Reviewer 1 Report

Thank-you for the opportunity to review this manuscript. This topic, how nurses have adapted professionally while caring for patients during the COVID-19 pandemic, is of high interest to readers and I commend the authors for exploring this topic.  As written, this manuscript requires significant edits.  The background section does not adequately set the stage for the significance of this work nor does it make the gap clear about why this study is needed. Furthermore, there are statements that read as causal relationships rather an observation (ex. line 37-38).  There are other statements that are simply not clear; for example line 32 suggests the COVID-19 is transmitted via contact (which is incorrect; it is airborne) and line 43 suggests that nurses work 24 hour shifts.  The research question does not align with the methods; experiences are explored using phenomenology, and ground theory should be focused on processes.  Furthermore, the research question should be revised to align with the methods. To set up the study clear aims helps to orient the reader to the study; the aims written here  (lines 78-83) are confusing and poorly written (what is basic data? and why is this one study aiming to influence policy?).  Without a clear background and significance, clear research questions and aims,  it is difficult to further assess the findings of this study. However, as it is currently written the results and the sheer number of categories and subcategories suggests the need for further abstraction of the data to assure complete analysis. The proposed model / theory is intriguing; however, the data and the manuscript (as currently written) is not presented in a way that allows the reader to assure adequate rigor and trustworthiness.  

Author Response

Dear Reviewer

We would like to thank you for your time and constructive feedback improving our manuscript, “Nurses’ Adaptation to Caring for COVID-19 Patients: A Grounded Theory Study” (ijerph-1198322), which we submitted to the International Journal of Environmental Research and Public Health. Please see our responses to the reviewers’ comments after this letter. Please see the attachment.

We hope that the changes we made have significantly improved the quality of our manuscript. Please do let us know if there is anything else we can do at this stage. We would be more than happy to do everything we can to assist you in this process.

We look forward to hearing from you!

Best Regards,

Suhyun Kim

Department of Nursing, Nambu University

23 Cheomdanjoongang-ro, Gwangsan-gu

Gwangju 62271, Republic of Korea

Tel.:82-62-970-0249; Fax:82-62-970-0261

E-mail: ksh136112@gmail.com

Reviewer 2 Report

Thank you very much for possibility to read this manuscript. This is important work, well planned, structured and written, with important implications for nursing practice.

I have only few issues which I recommend to consider:

  • in a discussion, when reading it is mainly repetition of findings. I would suggest to -re-write discussion to show mostly practical aspects coming from the findings.
  • in limitation - as it is qualitative study, I suggest to consider to add the risk of bias coming from analysis and interpretation.
  • in conclusion - repetition of the essence of the study at the begining is not necessary as it is underlined several times in the manuscript. It is better to focus on implication coming from the findings

Author Response

(The authors gave the same response as above.)

Round 2

Reviewer 1 Report

This study is presented as a grounded theory study which aims to "develop a substantive theory by exploring nurses' adaptation process in caring for COVID-19 patients and examining how nurses interact with the phenomenon and provide the basic data necessary to develop intervention programs and policies to support nurses caring for COVID-19 patients."  The research question presented is "What is the nurses' adaptation process in caring for COVID-19 patients?" This paper is timely, relevant, and potentially impactful contribution to nursing. However, as the manuscript is currently written, it lacks rigor, trustworthiness, and interpretability which undermines its contribute to science.

In a well written manuscript, each section of the paper flows into the next seamlessly. However, in this paper, this flow is not present. This is true throughout the entire paper from background to conclusion.

The background and significance sections, which  focuses on the psychological and physical consequences of caring for patients during a pandemic, does not align with the aim of "exploring nurses' adaptation process in caring for COVID-19 patients."  The background suggests that (line 55-56) the study will focus on those who are experiencing stress and exhaustion and thinking of leaving nursing; however none of these are mentioned in the inclusion criteria for the study. Furthermore, the adaptation process is never defined, nor does the background section indicate why this process should be explored, the implications of this process on nursing practice, or what is currently known about nurses adapting their practice.  As written, it is my understanding that the authors are seeking to understand what nurses do to adapt their nursing practice to care for patients with COVID-19. 

However, the methods section describes that the main question posed during the interviews was "Please tell me about your experiences of caring for COVID-19 patients." The results section then goes on to describe the experiences that nurses had while caring for COVID-19 patients, which does not achieve the aim that the researchers indicated.  What is reported on are the experiences of caring for these patients, with very little description of the adaptation process.  Furthermore, the authors seems to have coded their data to the dimensions and properties rather than coding their data to describe the phenomenon of interest: the adaption process of nurses to care for patients with COVID-19.  Additionally, the sheer number of categories and subcategories indicates a need to further abstract the data (further data analysis) to better represent the data. 

The results section of this manuscript suggests that the authors captured rich data from their participants; however, the data presented with the interpretation of those data do not align nor with the method of ground theory. For example, the subcategory of "mental burnout" (line 248) speaks to a myriad of concepts from "the burden of new tasks" to nurses feeling like novices.  The data presented in the excerpt from Participant 18 speaks to not knowing how to cope and lack of knowledge about the situation - which is quite different and has little to do with the established concept of burnout.  A second example is the data excerpts provided by Participants 18 and 19 (lines 453 -462) within the category of "growing stronger to get back up" the quotes indicate strategies used to either physically get stronger or to identify new processes for their practice but none of them speak to "getting back up" and it is also not described in relation to the process of adaption to care for patients with COVID-19.

Many of the categories and subcategories do not have any relevance in relation to the adaption process. In grounded theory the conditions (listed in this manuscript as the experiences and categories) should be described in relation to how they influence the adaptation process caring for COVID-19 patients (the phenomenon of interest). Throughout the results section, the reader should be guided through how the adaptation process occurs, under what conditions, and with what dimensions, rather than read a litany of experiences of nurses (as that does not align with the aim of the study nor the method).  The last section of the results section (section 3.3.6, line 532) speaks to the process (or outcome of the process); however, the authors have not led the reader to clearly understand what the process is, why it is important, or how the nursing practice changed or was adapted to care for these patients with COVID-19. The "adaptation process" (section 3.4, line 567) should be the focus of the paper with the conditions/categories and properties describing how the adaptation process is influenced by all of those other things. 

The conclusions do not align with any of the data presented or the findings reported. The implications for education should not be how this paper concludes. 

I recommend that the authors work with a mentor who is an expert in grounded theory to redirect their data analysis and presentation of the data.  I also would recommend that the manuscript be re-written to align each section and element of paper.  The aim should be clear and unambiguous.  In the re-writing of this paper, the authors should consider being transparent about their role in the study, how they are related to the study participants, and how their lens influences the data analysis and interpretation of the findings. Being transparent about their role may also help to understand the strong statements about the lack of resources provided during the pandemic (i.e. lines 619 - 624).

Finally, there were some ethical concerns that I noted which should be contextualized or omitted. For example, nurses be prevented from taking days off (line 281-283) and sharing of patient information broadly (lines 308-312). There are items within the paper that are presented factually; however, they aren't likely fact.  For example, line 315, to my knowledge, COVID is not determined with blood tests. Another example is lines 204 - 207 - how was it determined that nurses were burned out physically and psychologically? Or in line 234-235, how was it determined that the nurses practiced social distancing more than the general public?  Or in line 278, it is noted that nurses reported hypoxia, which is an objective measure, how were nurses measuring their oxygen level while wearing PPE and caring for patients? These types of details influence the rigor and trustworthiness of this paper.  

The authors do seem to have rich data and could contribute to what is known about nursing practice during a pandemic; however, to make a meaningful contribution to the science, it must be re-written with a refined presentation of further analyzed/re-analyzed data.